# Predicting the habitat suitability of *Ilex verticillata* (Aquifoliaceae) in China with field-test validations

Yawen Yin[1,2], Zhaobin Hou[3], Qiuyue Sun[1], Bailing Zhu[1], Jiaqi Liu[1], Yiping Zou[1,2], Mingzhuo Hao[1,2] *

1 College of Forestry and Grassland Science, College of Soil and Water Conservation, Nanjing Forestry University, Nanjing, Jiangsu, China, 2 Jiangsu Qinghao Landscape Horticulture Co., Ltd., Nanjing, Jiangsu, China, 3 Nanjing Police University, Nanjing, Jiangsu, China

* hmz@njfu.edu.cn

**Data Availability Statement:** The data underlying the results presented in the study are available from Zenodo (DOI: https://doi.org/10.5281/zenodo.10388329).

## Abstract

The cut branches of *Ilex verticillata* are highly ornamental and have high economic value. Since its introduction to China, it has received widespread attention. In the context of climate change today, ensuring its promotion and sustainable production in China is of great significance. In this study we evaluated the habitat suitability of the species using MaxEnt, combined with climate and soil variables, to assess the impact of climate change on its potential suitable habitat. We used 121 *I. verticillata* occurrence data and validated the model prediction using extensive field testing (12 test sites located in areas from 23.19˚ N to 42.91˚ N and 76.17˚ E to 125.14˚ E). The habitat suitability model (AUC = 0.854) performed well. Among them, three precipitation variables and one temperature variable were the main factors determining the distribution of *I. verticillata* in China. Field trial tests and model predictions of the suitability of *I. verticillata* were consistent, indicating that our model predictions are biologically meaningful and economically valuable. Under the Shared Socioeconomic Pathways (SSPs) climate change scenario, the high and medium suitable habitats for this species will be reduced in the future climate. This study helps to better understand the impact of climate change on *I. verticillata* and provides suggestions for the introduction and cultivation areas and protection of this species in China.

## 1.Introduction

*Ilex verticillata* (L.) A.Gray (Aquifoliaceae) is a perennial shrub of the holly genus, native to the northeastern United States. Currently, it is widely distributed in eastern Canada, northeastern and southern United States, France, China, and various European countries. Primarily utilized as potted plants and cut branches [1,2], *I. verticillata* stands out as a rare deciduous species within the Aquifoliaceae family. Noteworthy for its vibrant red and dense fruits, extended fruiting cycle, and heightened ecological and ornamental value [3]. This species was introduced to China in 1997, and large-scale introductions for breeding began in 2006 with

**Funding:** This research was supported by the Modern Agriculture Project of Jiangsu Province under Grant No. BE2021307, the Science and Technology Development Plan Project of Nanjing City, under Grant No. 202306003, the Jiangsu Provincial Innovation and Extension Project of Agricultural Science and Technology, under Grant No. 2023-SJ-009, and the Jiangsu Provincial Innovation and Extension Project of Forestry Science and Technology, under Grant No. LYKJ [2021]07.

**Competing interests:** The authors have declared that no competing interests exist.

multiple locations [4]. Presently, *I. verticillata* is gaining popularity in garden and indoor ornamental applications, emerging as a significant source of income in certain regions.

Since the 21st century, due to the impact of the greenhouse effect, various climate change factors have synergistically caused climate changes in some regions. Drought, flooding, overheating, and extreme cold, may negatively impact the yield and quality of *I. verticillata*, thereby affecting the economic benefits of related industries, affecting the economic traits of its cuttings [5]. Climate change has already exerted pressure on the plantation industry [6], so aiding *I. verticillata* to effectively cope with climatic variations is of utmost significance to plantation managers. In its original habitat, *I. verticillata* predominantly thrives in low-latitude wetlands such as marshes, wetlands, ponds and lakesides, exhibiting underdeveloped main roots and abundant adventitious roots—a characteristic of shallow-rooted species [7]. However, introduction and cultivation experiments have revealed that *I. verticillata* can thrive not only in wetland environments but also in general soil conditions. According to Chinese introduction results, *I. verticillata* planted in terrestrial environments manifests a more developed root system and enhanced growth potential [8]. As a result, *I. verticillata* has excellent adaptive capacity to cope with gradual climate change and to mitigate its negative impacts. As the cultivation of *I. verticillata* plantations intensifies, there is an urgent need for comprehensive assessments and detailed quantification to understand the potential risks of climate change on the reduction of suitable habitats for this species. This includes identifying and understanding potential threats through in-depth analysis and scientific methods, to minimize economic losses and ensure sustainable development in the face of changing environmental conditions. This will provide strong support for the development of effective conservation and adaptation strategies.

In recent years, bioclimatic models have been widely used to assess the suitability of species habitat under climate change. These models typically use the relationship between recorded species distributions and climatic variables to predict potentially suitable habitat for specific species [9]. For example, these methods have been used to predict the potential suitable climatic habitats of plants such as *Hydrangea macrophylla*, *Ginkgo biloba*, and *Metasequoia glyptostroboides* within China [10–13]. Although there are many bioclimatic models, many studies have shown that the maximum entropy (MaxEnt) typically provides effective discrimination within the modeling area and is widely used [14]. MaxEnt, in comparison to other models, is capable of providing more accurate predictions even when there is limited background data on species occurrence and environmental information across the entire study area. This algorithm has the ability to yield optimal ranges for each variable concerning the species and directly generate habitat suitability maps with explicit spatial information [15]. MaxEnt models, grounded in the input indicators, integrate an understanding of both the habitat of species and plant physiology. Consequently, the results derived from the model under different scenarios are more detailed and reliable.

It is worth noting that despite the rapid increase in the literature on bioclimatic modelling, only a few articles have validated maxent prediction results in the Chinese region. Climatic, soil and biological factors all influence the ability of shrubs to adapt to their environment. From a physiological point of view, shrubs can adapt to a globally warming climate by changing their phenotypic characteristics [16]. *I.verticillata* is a representative fruit-bearing plant with important ecological and ornamental value, as its fruits are used for the production of cuttings and as winter food for birds. The fruit traits of *I. verticillata* are directly related to its economic value, and different climatic and soil conditions have different effects on fruit traits [17]. There has been a lack of research on the geographic regional scale and habitat suitability of *I. verticillata*, which has resulted in loss of economic value due to inappropriate cultivation practices of *I. verticillata*.

This study aims to fill these gaps and provide guidance on the impacts of climate change on *I. verticillata* cultivation and climate-smart cultivation practices at the geographic scale. We studied the distribution of *I. verticillata* cultivation in China and found that its actual cultivation area is relatively scattered. In order to verify the differences in economic traits of *I. verticillata* grown under different climatic conditions, we selected three sites in each of the predicted unsuitable, low-suitable, medium-suitable, and high-suitable zones for field experiments. The response of *I. verticillata* to climate change was assessed by measuring two fruit phenotypic indicators, fruit brightness and fruit diameter. The reliability of the bioclimatic model was verified by predicting changes in fruit traits in response to climate in different ecological regions.

## 2. Materials and methods

### 2.1. Occurrence data

The distribution data of *I. verticillata* mainly comes from specimens. Coordinates were extracted from the Chinese Virtual Herbarium(http://www.cvh.ac.cn/), the Teaching specimen resource sharing platform(http://mnh.scu.edu.cn/), the Chinese Natural History Museum (http://www.nature-museum.net), the Global Biodiversity Information Facility (https://www.gbif.org/) [18], and other digital platforms. In addition, natural germplasm resources data from forestry bureaus, forestry stations, and university research institutions throughout the country were collected, and relevant journal articles and thesis published in China were searched and consulted for additional coordinates using Google Earth to supplement the accurate coordinates of the occurrence records. To minimize the clustering effect error in predicting potential distribution areas and avoid model overfitting, ENMTools (https://github.com/danlwarren/ENMTools) were used to screen and eliminate specimens with missing coordinates and duplicate data [19,20], and artificially planted samples were additionally manually removed. A single sample record was retained within a grid range of 2.5× 2.5 arcmin. Finally, we selected 121 distribution points for further analysis. The distribution points are shown in Fig 1.

The map of China was obtained from the National Mapping Geographic Information Bureau Standard Map Service website (http://bzdt.ch.mnr.gov.cn/index.html), using the China map with review number GS (2019) 1822 and a scale of 1:20 million (http://bzdt.ch.mnr.gov.cn/), which was converted to shapefile (shp) format for subsequent analysis and mapping.

### 2.2. Environmental data acquisition

Climatic variables have shown to have important ecological significance for plants, affecting their distribution at broader scales. In this study, climatic data were downloaded from World-Clim 2.1(1970–2000) (http://www.worldclim.org/), including 19 climate variables (Bio 1-Bio 19) and one elevation data, with a spatial resolution of 2.5 arcmin. Future climate scenarios were selected from the sixth international coupled model comparison project (CMIP6) Beijing climate center climate system model (BCC-CSM2-MR model) [21]. *I. verticilata* was introduced to China on a large scale after the 2000s, and the CMIP6 version of the data was used due to the relatively small climate change in China in the last 20 years. To further consider the impact of economic development patterns on climate change, this study selected shared socio-economic pathways (SSPs) which reflect the relationship between socio-economic development patterns and climate change risk [22]. Among them, SSP1-2.6 (minimum greenhouse gas emission scenario) is a sustainable green development path that belongs to a low greenhouse gas emission climate scenario, limiting global warming to below 2°C by 2100; SSP2-4.5 (medium greenhouse gas emission scenario) belongs to a medium greenhouse gas emission

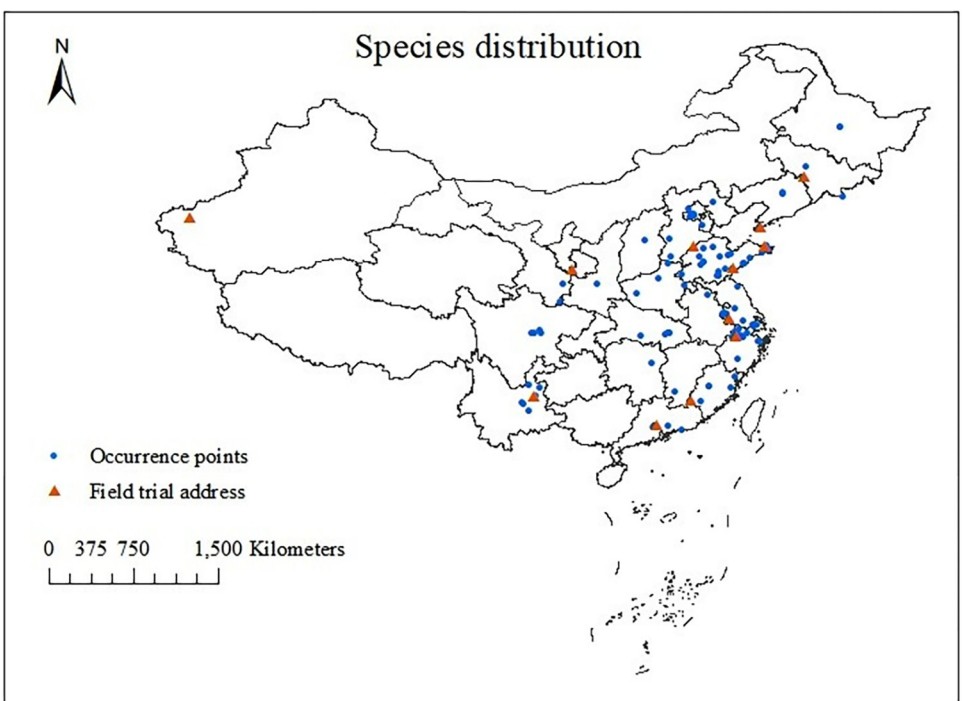

**Fig 1. The Chinese distribution points of *I. verticillata*.** Note: The blue circle icon represents the actual distribution point of *I. verticillata*. Red triangles represent test sites. This map is based on the standard map with review number GS (2019) 1822 downloaded from the National Bureau of Surveying, Mapping and Geoinformation's Standard Map Service website (http://bzdt.nasg.gov.cn/index.jsp, http://bzdt.ch.mnr.gov.cn/), and no modifications have been made to the base map. The base map is unmodified.

climate scenario, indicating that global temperatures will rise about 3˚C in the future; SSP5-8.5 (maximum greenhouse gas emission scenario) is a high population growth and high fossil fuel consumption climate scenario, belonging to the worst climate scenario for greenhouse gas emissions in the future [23,24]. Three scenarios (SSP1-2.6, SSP2-4.5, SSP5-8.5) and three periods (2050s, 2070s, 2090s) were selected as future environmental data sources. Eighteen topsoil indicators were obtained from the world soil database (HWSD, http://www.iiasa.ac.at/web/home/research/ researchPrograms/water/HWSD.html). Slope and aspect data were extracted from DEM data(2.5 arcmin) (source: http://www.gscloud.cn). To avoid model overfitting, we imported environmental factors and distribution point data into the MaxEnt model for pre-simulation, using the jackknife test to remove variables with a contribution rate of 0. We extracted species occurrence point information using ArcGIS for Pearson correlation analysis to measure autocorrelation between variables (R, version 4.3.1) (Fig 2). For two environmental factors with a Pearson correlation coefficient of $|r| \geq 0.8$, combined with the pre-simulation results, only environmental variables with high contribution rates and significant biological significance were retained in the model to eliminate multicollinearity between variables [25,26].

## 2.3. Model construction and optimization

Using the 'sdm' package in R, we considered four species distribution models (SDMs) to predict potential suitable distribution areas for *I. verticillata* in China: the generalised linear model (GLM), the BIOCLIM model, the maximum entropy model (MaxEnt), and the random forest model (RF). Model accuracy was assessed using a joint test of true skill statistic values

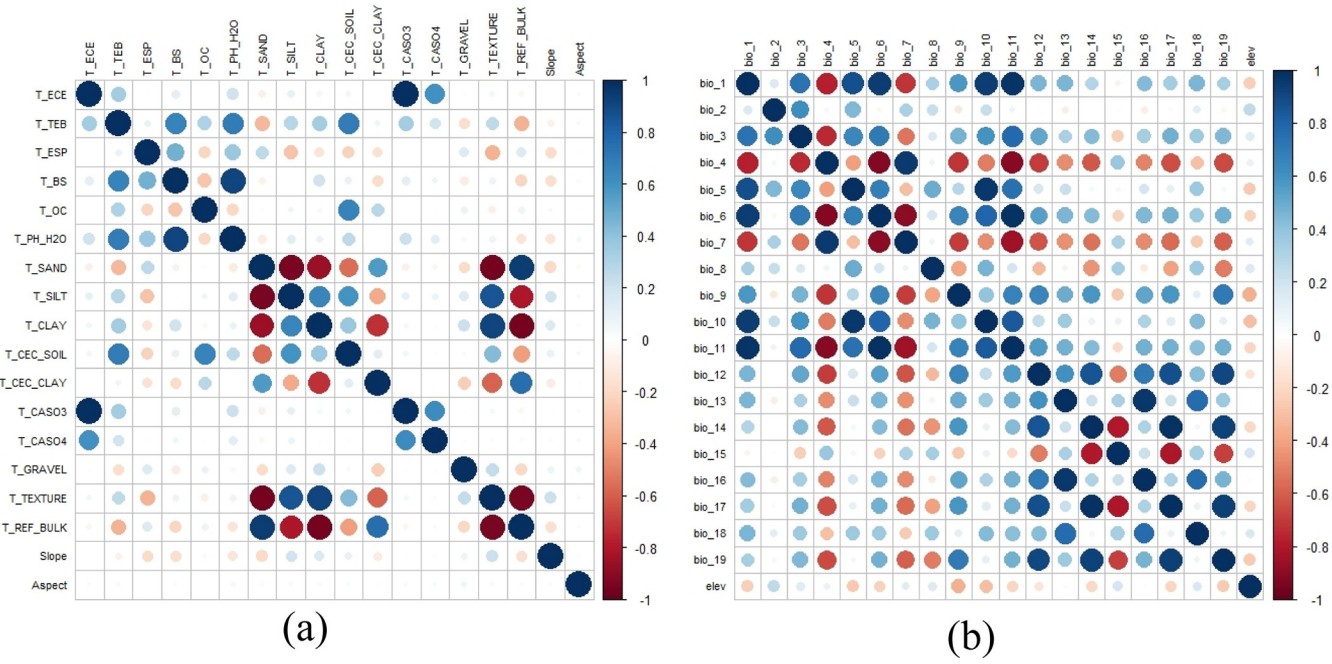

**Fig 2.** (a) Heat map of the correlation of environmental variables. (b) Heat map of the correlation of soil variables. The blue circles are for positive correlations and the red ones for negative correlations. The stronger the correlation, the larger the size of the circle and the darker the color.

(TSS) and subjects' work characteristics under the curve (ROC) [27]. TSS is a true-false-positive test and is not affected by the size of the validation data, which can make up for the shortcomings of Kappa in the assessment process [28]. When the area under the curve (AUC) of the ROC curve is less than 0.6, it indicates that the model is not valid, $0.6 \leq \text{AUC} < 0.7$ indicates that the model is fair, $0.7 \leq \text{AUC} < 0.8$ indicates that the model is moderate, $0.8 \leq \text{AUC} < 0.9$ indicates that the model is good, and $\text{AUC} \geq 0.9$ indicates that the model is excellent [29].

We optimized the MaxEnt 3.4.4 model to accurately simulate the relationship between species occurrence and climate variables. The prediction performance of the model is mainly affected by two parameters, the regularization multiplier (RM) and feature combination (FC). We set the RM constant to 0.5 to 4.0 with a step size of 0.1. For the selection of FC, we tested 29 combinations of five feature classes (linear = l, quadratic = q, product = p, threshold = t, hinge = h) [30]. We used the kuenm package in R to optimize the MaxEnt model [31]. In the optimization process, we randomly divided the data into a training set (75%) for model development and a validation set (25%) for model performance evaluation. To consider the uncertainty introduced by splitting the training and validation sets, we constructed 10 models for cross-validation by running 10 times. The remaining settings were kept as default. This setting has been considered reasonable and effective in extensive niche research. We tested the above 1161 candidate models and selected the best model based on the following criteria: (1) significance model, (2) omission rate $\leq 5\%$, and the model with the minimum AICc (Akaike information criterion) value (i.e., $\Delta$ AICc = 0) was considered the final model [32].

The built-in model quality assessment routine of MaxEnt was used to evaluate the accuracy of model predictions. First, through ten iterations, we obtained the standard deviation between models to assess possible deviations due to data partitioning. The area under the receiver operating characteristic curve (AUC) is a comprehensive index reflecting both model sensitivity and specificity [33], and in practical applications, the AUC index derived from ROC is

commonly used to evaluate model performance [34]. As the AUC is not affected by thresholds, it is a more reasonable basis for comparing different models. The optimized model had an AUC value of 0.854, which belongs to a good level and is acceptable for simulation [35]. In addition, AICc values and differences between training and testing AUC values [AUC. Diff] comprehensively reflects the goodness of fit and complexity of a model, both of which are excellent criteria for assessing model performance.

## 2.4. Model validation

To verify the accuracy of model predictions, we investigated *I. verticillata* fruits from 12 field trial sites across the country. Sampling was conducted from three suitable habitats (high, medium, and low) at each of the 12 planting sites. The distribution of test sites is labelled in the map of China in Fig 1, supplemented by specific information in Table 1. A small number of stunted fruits were collected from unsuitable areas where most of the plants exhibited fruit-lessness, low fruit yield and low fruit shape index. Due to the possible influence of tree age and genetic relatedness on *I. verticillata* fruit traits [36], we selected 5-year-old shrub type *I. verticillata* 'Oosterwijk '(provided by the Bai Ma Teaching Base of Nanjing Forestry University). Sampling trees were at least 60 m apart. We collected 100 seeds from 10 trees at sampling points in the four directions (east, west, south, north) at each planting base. Then, the fruits collected from each site were mixed uniformly and brought back to the laboratory for phenotypic index measurement. We selected 100 seeds from each sample and measured their diameter and fruit brightness. Between November and December of 2022, the relevant indicators of *I. verticillata* fruits were measured, and the measurement time was adjusted according to the fruit maturity status in each region. A handheld chromometer (Shenzhen 3nh Technology Ltd. Co., China) was used to measure fruit brightness, with the aim of assessing fruit coloration. A vernier caliper was used to measure fruit diameter. Fruit trait data were analysed and plotted using R 4.3.1 software. A one-way ANOVA was executed to compare the overall differences and P-value test was performed to analyse whether the effect of the factors on the results was significant or not. Multiple comparisons (Tukey HSD) were then performed to see two-by-two differences, with non-significant differences indicated by the same letter and significant differences indicated by different letters.

**Table 1. Geographical distribution and climate data of 12 test sites.**

| Category | Site | Latitude (°N) | Longitude (°E) | Altitude (m) | Mean Temperature(°C) | Mean precipitation(mm) |
|---|---|---|---|---|---|---|
| Unsuitable habitat | 1 | 39.72 | 76.17 | 1245 | 13.2 | 125.5 |
| | 2 | 42.91 | 125.14 | 379 | 6.3 | 765.9 |
| | 3 | 35.55 | 106.67 | 1398 | 9.5 | 511.2 |
| Low-suitable habitat | 1 | 37.44 | 116.37 | 79 | 12.9 | 547.5 |
| | 2 | 38.92 | 121.62 | 20 | 10.5 | 662.3 |
| | 3 | 23.19 | 113.37 | 64 | 24.5 | 1781.8 |
| Medium-suitable habitat | 1 | 30.24 | 119.74 | 187 | 18.1 | 1613.9 |
| | 2 | 25.13 | 116.14 | 274 | 20.6 | 1780.5 |
| | 3 | 31.59 | 119.19 | 28 | 16.4 | 1204.3 |
| High-suitable habitat | 1 | 35.68 | 119.44 | 69 | 14.1 | 813 |
| | 2 | 37.41 | 121.99 | 61.3 | 12.9 | 730.2 |
| | 3 | 25.43 | 103.59 | 1871 | 13.6 | 999.7 |

## 2.5. Prediction of climate change impact and suitable habitat division

We projected the spatial distribution of suitable habitats in China for the current period (2020) and the next three periods (2050, 2070, and 2090) and selected three climate change scenarios (SSP1-2.6, SSP2-4.5, and SSP5-8.5) [37]. Then, we compared the changes in area and spatial range of these predictions to the reference period for different categories of suitable habitats. MaxEnt assigned non-negative probabilities (with a total probability of 1) to each pixel in the study area. To facilitate ease of use and interpretation, MaxEnt outputs the "cumulative" probability of each pixel in a 0–100% range. Therefore, habitat suitability was classified by setting a decision threshold. Based on previous research, we divided habitat suitability into four levels using natural breaks: (1) high habitat suitability with a probability value greater than 0.6; (2) medium habitat suitability with a value between 0.4 and 0.6; (3) low habitat suitability with a value between 0.1 and 0.4; and (4) unsuitable habitat with a value less than 0.1 [38]. The specific flow chart is shown in Fig 3.

## 3.Results

### 3.1 Accuracy assessment and selection of best prediction models

Based on the correlation analysis among environmental factors, PCA loading values and the contribution to the model in the pre-experiment, 15 environmental factors with correlations less than 0.8 and with large contributions to the model were finally retained for modelling, including 10 climatic factors, 1 topographic factor and 4 soil factors (Table 2). Before formal

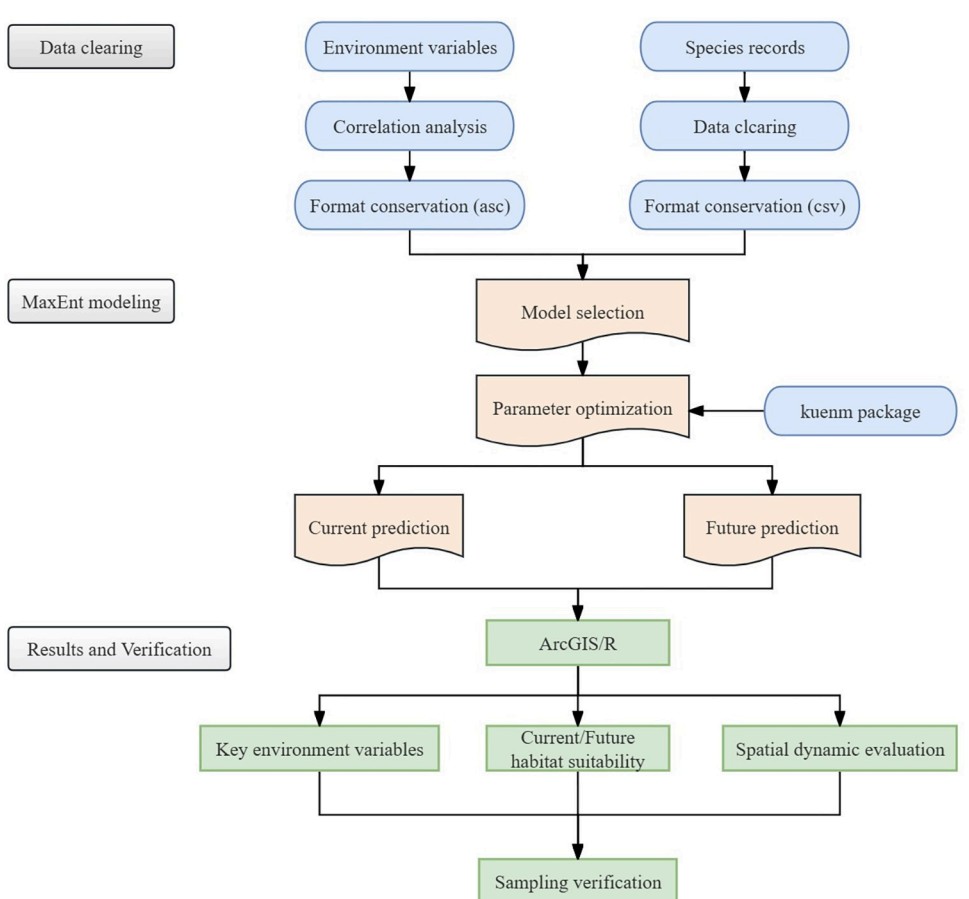

**Fig 3. Analytical framework for predicting potential *I. verticillata* distributions.**

**Table 2. Environmental factors selected through correlation analysis and model construction.**

| Data type | Variable code | Climatic factors | Units | Percent contribution |
|---|---|---|---|---|
| Climate factor | Bio1 | Annual mean temperature | (˚C) | 2.4 |
| | Bio3 | Isothermality (bio2/bio7) (*100) | (–) | 0.6 |
| | Bio4 | Temperature seasonality (standard deviation×100) | (–) | 19.9 |
| | Bio7 | Temperature annual range (bio5 –bio6) | (˚C) | 6.7 |
| | Bio10 | Mean temperature of warmest quarter | (˚C) | 5.6 |
| | Bio12 | Annual precipitation | (mm) | 43.2 |
| | Bio14 | Precipitation of driest month | (mm) | 2 |
| | Bio15 | Precipitation seasonality(coefficient of variation) | (–) | 10.8 |
| | Bio17 | Precipitation of driest quarter | (mm) | 7.4 |
| | Bio19 | Precipitation of coldest quarter | (mm) | 0.3 |
| Topographic Variable | slope | Slope | (˚) | 0.2 |
| Soil factor | T_bs | Topsoil base saturation | (%) | 0.4 |
| | T_clay | Topsoil clay fraction | (%) | 0.2 |
| | T_sand | Topsoil sand fraction | (%) | 0.1 |
| | T_pH_$H_2O$ | Topsoil pH ($H_2O$) | (-log(H+)) | 0.1 |

modelling, the accuracy of several common types of models in the prediction process was first evaluated. Overall, except for BIOCLIM, the SDM models of GLM, MaxEnt and RF had good potentials in predicting the distribution of *I. verticillata*, with TSS values greater than 0.7 and AUC values over 0.8. The MaxEnt model outperformed the other three models, and was more reliable in predicting the distribution of *I. verticillata*. The MaxEnt model outperformed the other three models (Fig 4). After further optimisation based on the MaxEnt model, the average AUC of 10 replications was 0.854 with a standard deviation of 0.014, indicating that the model had higher accuracy (Table 3).

## 3.2. Contribution of climate variables

The results of jackknife test showed that the climate variable with the largest contribution to the model was bio12, accounting for 43.6%, followed by bio 4, accounting for 19.9%, bio15, accounting for 10.8%, and bio17, accounting for 7.4%. Among these four variables, three were precipitation variables and one was a temperature variable, explaining 81.7% of the model cumulatively. The contributions of the other variables to the model were relatively small. The jackknife test results showed that the prediction performance of all climate variables except slope was acceptable (AUC > 0.7). Response curves reflect the relationship between environmental variables and species suitability. The four climate variables at favorable levels (bio12, bio4, bio15, bio17) fell within the ranges of 700–1700 mm, 700–1250, 0–60, and 90–390 mm, respectively, indicating conditions more conducive to the survival of *I. verticillata*.

## 3.3. Distribution of current habitat suitability

The appropriateness of the habitat for *I. verticillata* in the present climate was depicted across four types (Fig 5). Predominantly, these suitability types were concentrated in the central and eastern regions. The region of high-suitability habitat was small, accounting for 1.94% of the country's land area. The medium-suitability habitat was widespread in the high-suitability habitat, accounting for 6.91%, and the low-suitability habitat accounts for 6.77%. Other areas were unsuitable habitats, accounting for 84.38%.

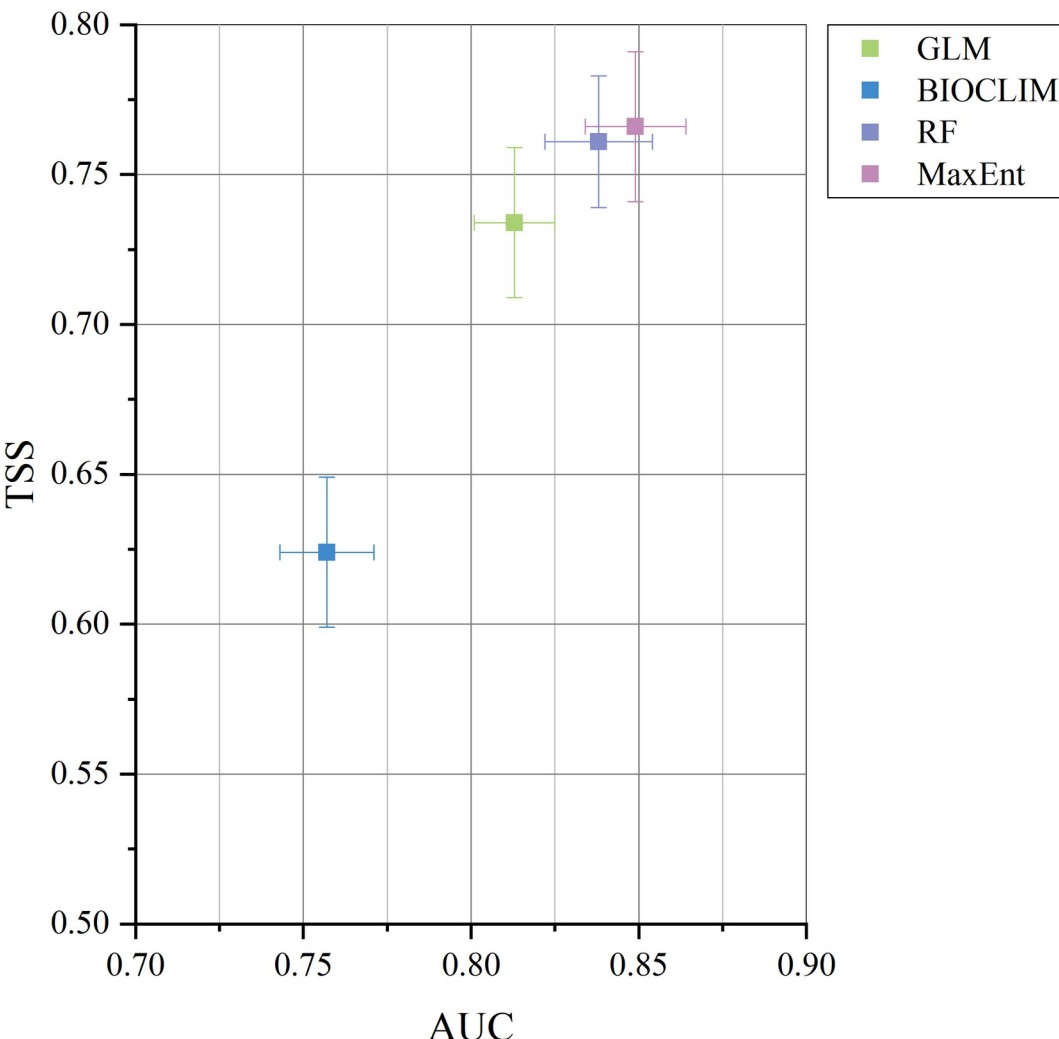

**Fig 4. Accuracy assessment of four models.**

### 3.4. Validation of bioclimatic model

There were significant differences in fruit brightness and size among optimal, suboptimal, marginal and unsuitable habitats of *I. verticillata* ($p < 0.05$). In different levels of habitat, These two traits demonstrate similar adaptive trends to the environment. The regional

**Table 3. Evaluation metrics of the default and optimal MaxEnt models by kuenm package.**

| Setting | FC | RM | AUC Diff | Omission rate | Delta AICc | AICc | AUC |
|---|---|---|---|---|---|---|---|
| Default | LQPH | 1 | 0.1895 | 0.0604 | 2649.982 | 102781.8 | 0.832 |
| Optimized | LPTH | 2.3 | 0.0008 | 0.0494 | 0 | 102760.7 | 0.854 |

FC: Feature combination,L(Linear features),Q(Quadratic features),P(Product features),H(Hinge features),T(Threshold features); RM:Regularization multiplier; AUC Diff.

Difference between the AUC values; Omission rate:Errors or performance degradation caused by data missing; Delta AICc:AICc The minimum information criterion AICc value; AICc:The akaike information criterion corrected; AUC:Area Under Curve.

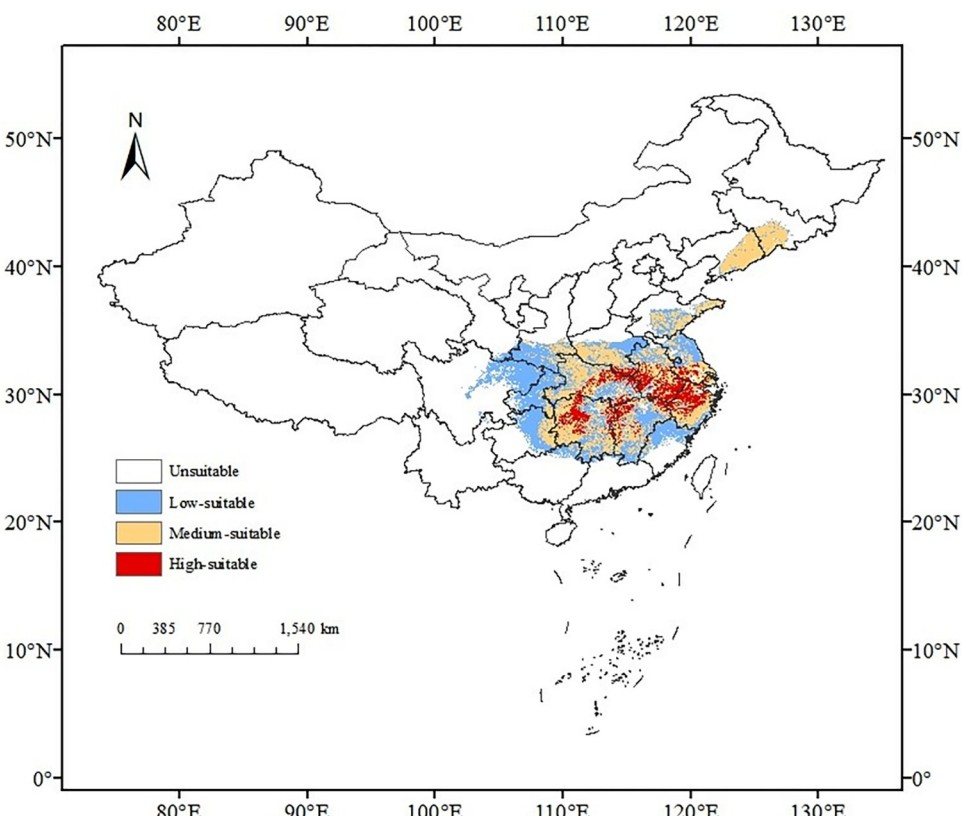

**Fig 5. Potential suitable regions for *I. verticillata* in China under current conditions.** Note: The map shows the current suitable areas for *I. verticillata* in China. The white areas indicate unsuitable areas, blue areas indicate low suitable areas, and red areas indicate high suitable areas. This map is based on the standard map with review number GS (2019) 1822 downloaded from the National Bureau of Surveying, Mapping and Geoinformation's Standard Map Service website http://bzdt.nasg.gov.cn/index.jsp, http://bzdt.ch.mnr.gov.cn/), and no modifications have been made to the base map. The base map is unmodified.

variations in phenotypic data, specifically fruit brightness and diameter, align closely with the rankings of habitat suitability classification. The experimental results validate the effectiveness of our model predictions. It suggested that the greater the environmental adaptability of *I. verticillata* fruit traits such as brightness and transverse diameter, the better the performance (Fig 6). *3.5 Prediction of changes in suitable habitat in the future period*

Utilizing the established bioclimatic model, the prediction of future suitable areas showed that the geographical distribution and area of *I. verticillata* habitat in China will undergo changes in the coming period (Fig 7). The suitable habitat will move northwestward, but the longitudinal displacement will not be significant. Under the SSP5-8.5 climate scenario, the displacement distance of the center of mass reached the furthest extent, with an expected shift of 150.47 km by 2090 (Fig 8). Compared with the current distribution, at the end of this century under the SSP5-8.5 scenario, the areas of high-quality, medium-quality, and low-quality suitable habitats will increase to 2.3%, 15.09%, and 6.6%, respectively. More significant changes will occur at the northwest edges of the three types of appropriate habitats.

## 3.6. Analysis of spatial pattern changes

Under the SSP1-2.6, SSP2-4.5, and SSP5-8.5 scenarios, the area of low, medium, and high suitability habitats is predicted to increase in the future, with a larger increase in medium

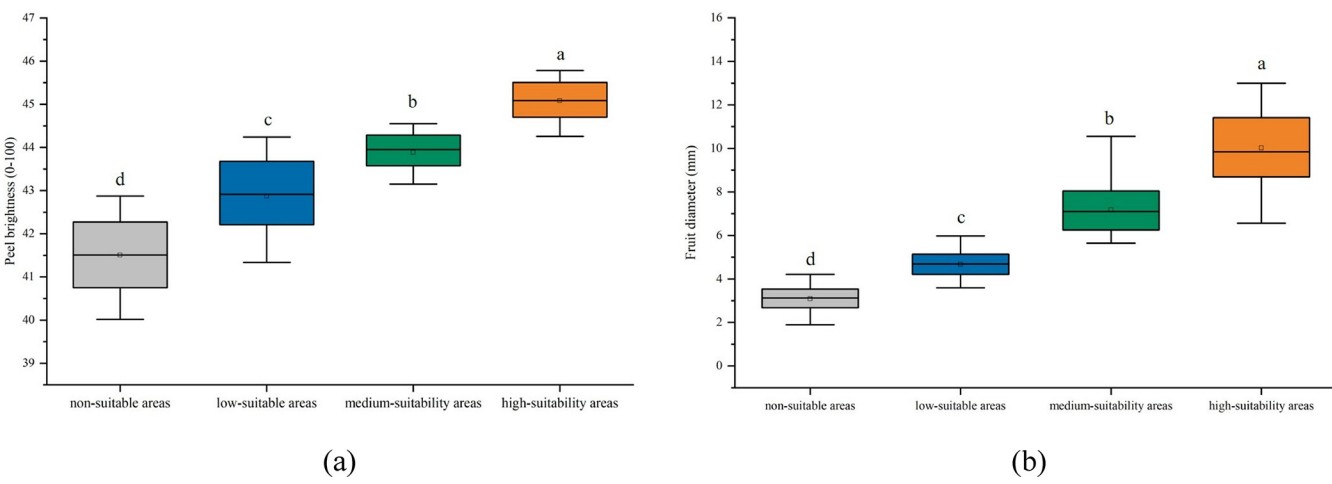

**Fig 6. Differences in *I. verticillata* fruit traits among suitable habitat classes observed based on one-way ANOVA.** Figures (a) and (b) show the differences in fruit brightness and fruit diameter (mm), respectively. Different letters indicate significant differences between means ($p < 0.05$).

suitability habitats (Fig 9). Under the SSP5-8.5 scenario, all projected changes are greater than those under the SSP1-2.6 scenario. However, under future climate conditions, the study area will become more suitable. Based on the graphs of changes in suitable area in Figs 9 and 10, under the SSP1-2.6, SSP2-4.5, and SSP5-8.5 scenarios, by the 2050s, highly appropriate habitat was expected to increase by 0.55%, 0.08%, and 0.96%, respectively, while moderately suitable habitat was expected to increase by 6.66%, 6.36%, and 8.64%, respectively. By the 2070s, highly suitable habitat was expected to increase by 2.62%, 0.75%, and 0.24%, respectively, while moderately suitable habitat was expected to increase by 4.93%, 6.68%, and 7.87%, respectively. By the 2090s, extremely appropriate environment was expected to have enlarged by 0.55%, 3.73%, and 0.37%, respectively, while moderately suitable habitat was expected to increase by 3.01%, 2.93%, and 8.18%, respectively. Most unsuitable areas and low-suitable habitats will be converted to medium and high-suitable habitats (Fig 10).

## 4. Discussion

*I. verticillata* is an important shrub species, both economically and ecologically speaking, but it faces great uncertainty under climate change. Understanding and quantifying the response of *I. verticillata* to rapid climate change is crucial. In this study, we used the MaxEnt model to predict the suitable habitat of *I. verticillata*, incorporating climate and soil variables. The final model obtained after optimisation (AUC = 0.854, SD = ±0.104) showed good overall performance. However, there were still some problems with using the MaxEnt model for prediction, and despite the inclusion of additional variables to improve reliability, there was still uncertainty in the results. Practical validation of growth metrics is still lacking as plant growth is affected by various biotic and abiotic factors such as interspecific competition, pests and animal damage [39,40]. Field data is a new method to verify the accuracy of SDM predictions, and we confirmed the model predictions through field trials. The results indicate that the *I. verticillata* industry in China is generally optimistic. Under three climate change scenarios, the area of high-suitable habitat and moderate-suitable habitat increased to varying degrees. Future predictions showed that with climate change, the favorable habitats for *I. verticillata* will expand their geographical range towards the northwest. Therefore, our research findings can serve as a prediction for action by seedling planting practitioners and policymakers, and provide a theoretical basis for the development of plantation planning.

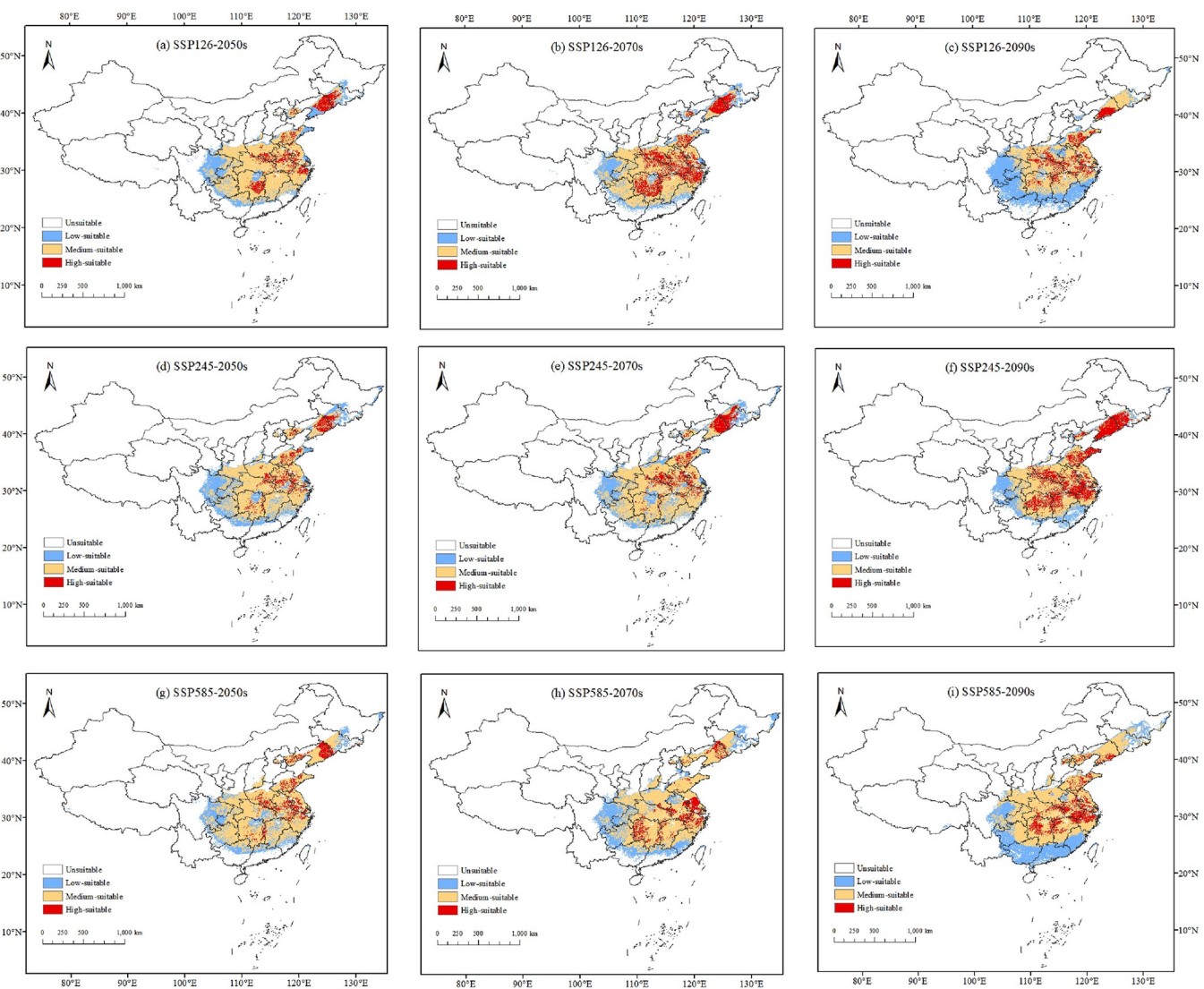

**Fig 7. Maps of habitat suitability for *I. verticillata* in China in 2050, 2070, and 2090 under emission scenarios SSP1-2.6 (a-c), SSP2-4.5 (d-f), and SSP5-8.5 (g-i).** Note: This map is based on the standard map with review number GS (2019) 1822 downloaded from the National Bureau of Surveying, Mapping and Geoinformation's Standard Map Service website (http://bzdt.nasg.gov.cn/index.jsp, http://bzdt.ch.mnr.gov.cn/), and no modifications have been made to the base map. The base map is unmodified.

### 4.1. Key climate factors determining the distribution of *I. verticillata*

The distribution of species is primarily driven by climate factors [41,42], but it is also limited by soil factors. Soil conditions affect temperature, humidity, precipitation, and processes of surface weathering and erosion [43]. The addition of soil variables to the model for prediction, especially under future climate scenarios, significantly reduced the total area predicted as suitable habitat [44]. When both climate and soil variables were included in the model for this prediction, a combination of these two categories was used for modelling. In China, favorable habitats are mainly distributed in the latitudes of 25˚-45˚N, with the most suitable growing conditions for *I. verticillata* in subtropical and temperate monsoon climate zones. We discovered that climate variables accounted for a significant proportion of the model's explanation (98.94%). The model in this study is based on the following assumption: the distribution

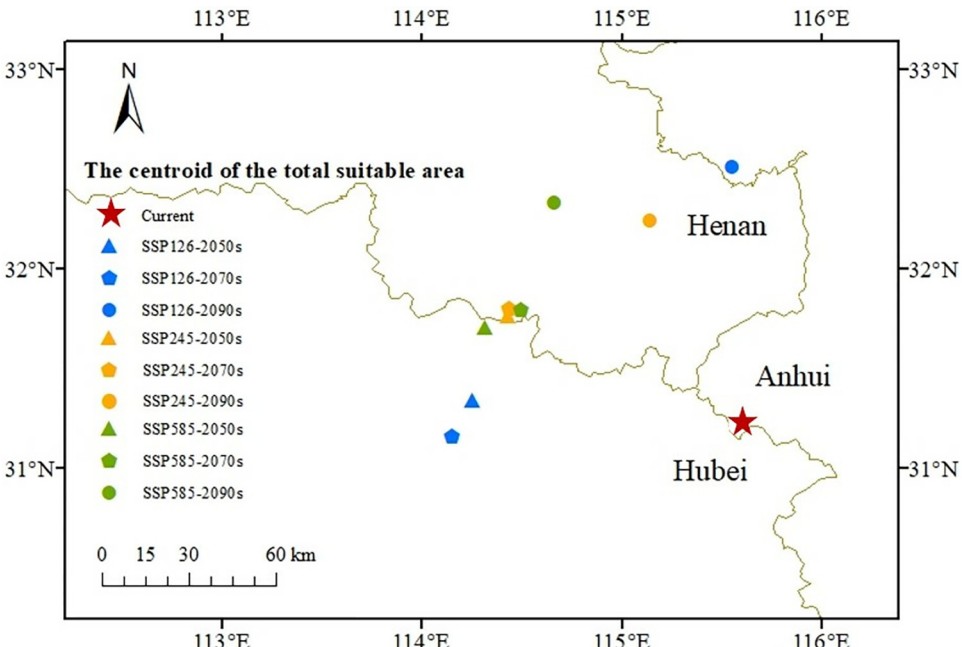

**Fig 8. The direction of the shift in the center of mass of the total suitable habitat of *I. verticillata* under three different climate scenarios (SSP1-2.6, SSP2-4.5, and SSP5-8.5) in the future.** Different climate scenarios are distinguished by distinct symbols; the current time is represented by a red star, while the 2050s, 2070s and 2090s are differentiated using specific colors. Note: This map is based on the standard map with review number GS (2019) 1822 downloaded from the National Bureau of Surveying, Mapping and Geoinformation's Standard Map Service website (http://bzdt.nasg.gov.cn/index.jsp, http://bzdt.ch.mnr.gov.cn/), and no modifications have been made to the base map. The base map is unmodified.

characteristics of *I. verticillata* within the land area are mainly determined by climatic factors. The simulation results in many articles support this assumption, which proves that, at a macroscopic level, the distribution changes of many species are consistent with the current changes in climate characteristics [45]. The top four climate indices contributing to the model were climate variable bio12 accounting for 43.6%, followed by bio4 accounting for 19.9%, bio15 accounting for 10.8%, and bio17 accounting for 7.4%. Among the influencing factors, there are three factors related to precipitation. Among them, bio12 ranks first in both the

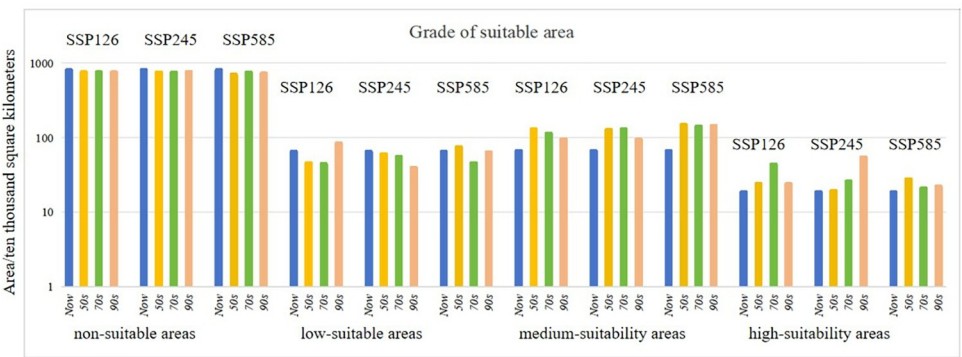

**Fig 9. Under three different climate scenarios (SSP1-2.6, SSP2-4.5, and SSP5-8.5) in three time periods-the 2050s, the 2070s, and the 2090s-the suitable habitat area of *I. verticillata* displayed distinct trends in change.** The four distinct suitability regions were easily distinguishable through color-coding.

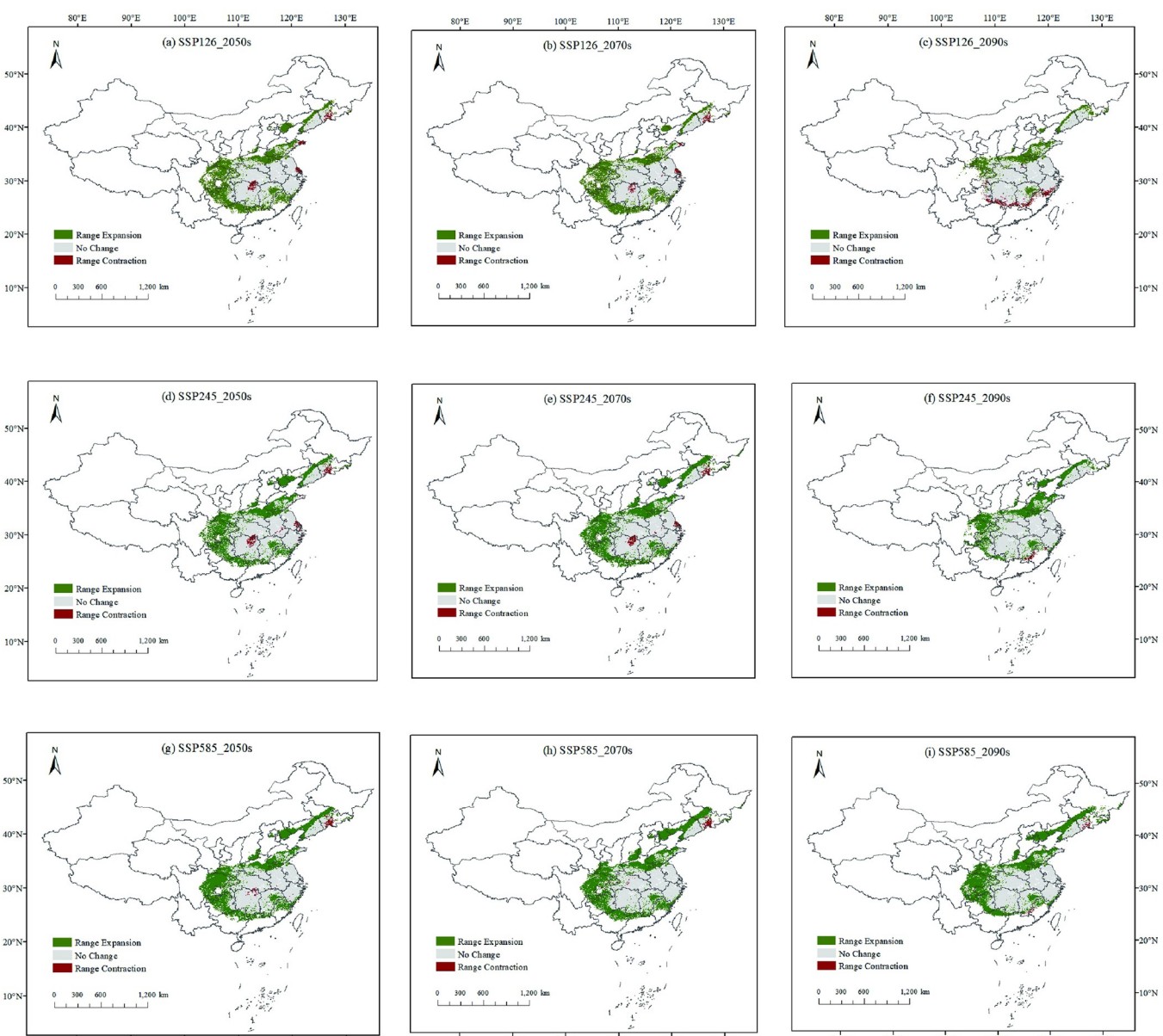

**Fig 10. Spatial pattern changes of *I. verticillata* under three different climate scenarios (SSP1-2.6, SSP2-4.5, and SSP5-8.5) in three time periods: 2050s, 2070s, and 2090s.** White represents unsuitable habitat areas, gray represents unchanged areas, green represents expanding areas, and red represents contracting areas. Note: This map is based on the standard map with review number GS (2019) 1822 downloaded from the National Bureau of Surveying, Mapping and Geoinformation's Standard Map Service website (http://bzdt.nasg.gov.cn/index.jsp, http://bzdt.ch.mnr.gov.cn/), and no modifications have been made to the base map. The base map is unmodified.

comprehensive contribution rate and independent prediction ability of the model, making it the most important factor limiting the distribution range of *I. verticillata*. The native habitat of *I. verticillata* is in swampy areas of North America, and it has a shallow root system with many fine roots. Water is crucial for the growth of *I. verticillata*. Water directly affects plant emergence and growth. Repeated exposure to drought and extreme rainfall conditions can profoundly impact the physiological and biochemical processes of plants by inducing changes in their biochemical characteristics, especially in terms of antioxidant activity, molecular changes,

chlorosis, and leaf necrosis [46–48]. The main source of water for field-growing *I. verticillata* is precipitation, and water management is also crucial for *I. verticillata*, which is considered the main factor determining whether *I. verticillata* can grow normally in a target area [49]. Bio4 is a temperature factor, and the current temperate continental climate, temperate monsoon climate, and subtropical monsoon climate zones are more suitable for the development of *I. verticillata*. During the growing season of *I. verticillata*, Sufficient rain and heat conditions provide necessary water and energy for plants, which is beneficial for the progress of photosynthesis, promoting the accumulation and increase of biomass, thereby accelerating the growth process of plants [50]. At the same time, winter low temperatures help to promote leaf defoliation and fruit coloration, increasing commercial quality and reducing disease threats [51,52]. Therefore, it is currently not advocated to establish large-scale *I. verticillata* plantations in areas with low annual precipitation and high temperatures. With climate change, it will be possible to introduce *I. verticillata* into currently low-rainfall areas in the future.

## 4.2. Model prediction

In this study, the three categories of climate suitability for the habitat of *I. verticillata* were distributed in the central and eastern regions of China (Fig 5). Under current climate conditions, the moderate and high suitability habitats of *I. verticillata* were mainly concentrated in the provinces of Hubei, Jiangxi, Hunan, Zhejiang, Jiangsu, and Shandong, as well as in the southern regions of Jilin and Liaoning and the southeastern region of Henan. The low suitability habitat was distributed in areas such as the eastern region of Guizhou Province, the southeastern region of Yunnan Province, and the northern region of Fujian Province. The majority of other regions in China were not suitable habitats for *I. verticillata* (accounting for 84.39%). Interestingly, the increasing trend of fruit brightness and fruit diameter of *I. verticillata* was consistent with the measured data changes in the four suitable categories (including unsuitable areas, low suitable areas, medium suitable areas, and high suitable areas). *I. verticillata* is an important ornamental fruit species that is often used for the production of cuttings. Fruit brightness and fruit diameter are key indicators for evaluating the growth of ornamental fruit plants. Precipitation and light have a significant impact on fruit brightness and fruit diameter [53,54], and there was little variation in these characteristics among individuals of the same species from the same site (6.7% and 26%, respectively). The predicted habitat suitability grades for *I. verticillata* using this model, combined with knowledge of plant ecology, physiology and economics, have important implications for guiding the selection of *I. verticillata* plantations in production practices.

## 4.3. Strategies for cultivating and conserving *I. verticillata* under climate change

We predict that under two future climate scenarios (SSP2-4.5 and SSP5-8.5), the high-suitability habitat for *I. verticillata* plantations will increase. Regions that are currently not suitable for *I. verticillata* growth, such as southern Gansu Province, southeast Shanxi Province, northeast Hebei Province, northern Guangdong and Guangxi Provinces, and Guizhou Province, will become suitable for *I. verticillata* growth. At the same time, under three future climate scenarios, the suitable habitat for *I. verticillata* will shift northwestward. The migration rate through natural seed dispersal is low, far lower than the rate needed to keep pace with expected changes in habitat [55]. Generally, there are many artificial *I. verticillata* plantations. In order to reduce the adverse effects of climate change on *I. verticillata*, we can refer to the predicted range and artificially introduce the species into areas outside its original distribution range. Clonal propagation and assisted gene breeding techniques can also improve its stress tolerance [56,57].

Therefore, plantation managers need to develop response plans that consider the transfer of appropriate habitats to maintain the quality and yield of *I. verticillata* cuttings in a changing climate.Based on a meta-analysis of long-term trends for over 1700 species, previous studies have shown that more than half of the species have experienced significant changes in distribution over the past 20 to 140 years [58,59]. Consequently, it is crucial to predict the long-term adaptability of *I. verticillata* under the climate circumstances of the future. Model predictions for three future periods indicate that suitable habitats (high and moderate suitability) for *I. verticillata* will increase and shift northwestward under climate change. These predictions are in line with previous studies that suggest climate change will initially affect the peripheral regions of species' distributions [60,61]. However, similar to many tree species, the suitability of habitats in northern China will benefit from climate change. Even though Heilongjiang Province is currently an inhospitable environment for *I. verticillata*, there may be moderate to low suitable habitats for *I. verticillata* in southeastern Heilongjiang Province in the future. Under three climate scenarios, the increase in the area of moderately high-suitability habitat will exceed the decrease during the three predicted periods.

The goal of this study is to construct a bioclimatic model to describe the suitable habitat of *I. verticillata* in China, in order to predict the potential impact of climate change on its environmental suitability. We identified the main climate components that affected the dissemination of *I. verticillata* and validated our predictions through field experiments. The final predicted results indicate that moderate climate warming may have a positive impact on the growth of *I. verticillata* within a certain period. These predictions can serve as a reference for plantation managers and germplasm resource conservationists to devise adaptation strategies in response to rapid climate change, enabling sustainable development of the plantation industry and protecting the germplasm resources of *I. verticillata*.

Although the distribution of *I. verticillata* is largely influenced by climatic and environmental factors, they also interact intricately with the surrounding biota through the establishment of Above- and below-ground animals, microorganisms, symbiotic partners, and resource-competing species all interact with them, so *I. verticillata* needs to actively regulate its own distribution to ensure access to optimal nutrients and ecological niches [62]. However, in *I. verticillata* plantations, human intervention and plantation management practices play a significant role in addition to natural conditions including factors like climate and soil. Hence, in future efforts to guide the cultivation and conservation of *I. verticillata*, it is essential to take into account additional factors that were not addressed in this study. The findings of this research provide a valuable foundation for future work, but further exploration is needed to address the broader ecological and environmental context in which *I. verticillata* exists.

## Acknowledgments

We would like to thank the research institutes and companies in various regions of China for their assistance during the field investigation and experiment. We also thank the anonymous reviewers and the handling editor for their valuable comments.

## Author Contributions

**Conceptualization:** Yawen Yin.

**Data curation:** Yawen Yin.

**Formal analysis:** Qiuyue Sun.

**Funding acquisition:** Qiuyue Sun.

**Investigation:** Bailing Zhu.

**Methodology:** Bailing Zhu.

**Project administration:** Jiaqi Liu.

**Resources:** Jiaqi Liu.

**Software:** Yiping Zou.

**Supervision:** Yiping Zou.

**Validation:** Mingzhuo Hao.

**Visualization:** Mingzhuo Hao.

**Writing – review & editing:** Zhaobin Hou.

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
