## [Decision Letter · Decision Letter 0]

16 Jun 2024

PONE-D-24-15741Predicting the Habitat Suitability of Ilex verticillata in China with Field-Test ValidationsPLOS ONE

Dear Dr. yin,

Thank you for submitting your manuscript to PLOS ONE. After careful consideration, we feel that it has merit but does not fully meet PLOS ONE’s publication criteria as it currently stands. Therefore, we invite you to submit a revised version of the manuscript that addresses the points raised during the review process.

I have received two expert assessments of your paper, and although I agree with the reviewers' opinions and suggestions, I understand that the paper still needs to adjust the methodology used. Note that this was one of the main points raised by the reviewers. Below I present the main criticisms regarding the methodology.

General comments after my reading:

-The use of MaxEnt is widely accepted; however, modeling algorithms are very diverse and can produce different predictions. This becomes particularly important for non-native species and for estimates considering future climate change scenarios. There is a vast scientific literature recommending the use of different algorithms (e.g., MaxEnt, BIOCLIM, RandomForest among others). Therefore, I recommend that the authors use various modeling techniques and produce a consensus map

-The species “*Ilex verticillata*” originates from the Northeast of the United States and has already been introduced in various parts of the world. This was briefly mentioned by the authors. The critical point is which points were used for modeling? Note that this was also not clear to reviewer 1. According to the text and figure 1, the authors used points from the native region (USA) and China. By doing this, the authors are assuming that the species' niche is in equilibrium. Another option is to use only the points from the native area and project the species' distribution in the invaded area. This approach assumes that the niche is conserved. There is a vast literature discussing the transferability of niche models for the case of invasive species, and the authors need to justify the methodology used.

-Please note that it is necessary to use the International Code of Nomenclature for algae, fungi, and plants when citing a plant species. For example, is necessary inform the taxonomic group (e.g. family) and author of the species.

Therefore, I invite the authors to review the manuscript and respond to the comments/suggestions presented.

We look forward to receiving your revised manuscript.

Kind regards,

João Carlos Nabout

Academic Editor

PLOS ONE

Journal Requirements:

"This research was funded by Jiangsu modern agricultural industrial technology system construction project. 

Grant No. JATS[2022]396"

6. We note that your Data Availability Statement is currently as follows: All relevant data are within the manuscript and its Supporting Information files.

7. When completing the data availability statement of the submission form, you indicated that you will make your data available on acceptance. We strongly recommend all authors decide on a data sharing plan before acceptance, as the process can be lengthy and hold up publication timelines. Please note that, though access restrictions are acceptable now, your entire data will need to be made freely accessible if your manuscript is accepted for publication. This policy applies to all data except where public deposition would breach compliance with the protocol approved by your research ethics board. If you are unable to adhere to our open data policy, please kindly revise your statement to explain your reasoning and we will seek the editor's input on an exemption. Please be assured that, once you have provided your new statement, the assessment of your exemption will not hold up the peer review process.

8. Please ensure that you refer to Figure 2, 7 and 8 in your text as, if accepted, production will need this reference to link the reader to the figure.

Reviewers' comments:

Reviewer's Responses to Questions

**Comments to the Author**

1. Is the manuscript technically sound, and do the data support the conclusions?

Reviewer #1: Partly

Reviewer #2: Yes

2. Has the statistical analysis been performed appropriately and rigorously? 

Reviewer #1: Yes

Reviewer #2: Yes

3. Have the authors made all data underlying the findings in their manuscript fully available?

Reviewer #1: Yes

Reviewer #2: Yes

4. Is the manuscript presented in an intelligible fashion and written in standard English?

Reviewer #1: No

Reviewer #2: Yes

5. Review Comments to the Author

Reviewer #1: The study by Yin et al. used a fashioned and improved method to evaluate the suitability of Ilex verticillata in China using MaxEnt, plus a nice evaluation in field using fitness related traits of fruits. My general impression is that the methodological section sounds logical but the overall manuscript will benefit from a better streamlining (specially a workflow figure at the beginning of the methods), and a consistency of terms. I have a major suggestion of how to improve part of the methods, specially the selection of suitability types, by using a comparison of an introduced model (China) vs a global one. However, I will understand if these changes are not totally feasible as you authors have already selected the 12 sites to test with field data. Other revisions are needed in typos, format, and discussion of soil variables. All in all, I had a great time reviewing the article so I will be glad of reviewing it again after the revisions.

Please see more detailed comments ahead:

Abstract

-line 22: MaxEntropy is also the software or type of algorithm, to be more consistent just says habitat suitability model as in line 26.

So, something like “In this study we evaluated the habitat suitability of the species using MaxEnt, combined with climate and…”

-line 26: “excellently” looks arbitrary and not too scientific. I would rewrite the sentence.

-line 30-31: I think you do not need to repeat “pathway”

Introduction

-line 37: include the letters for the scientific name author

-line 37: “originally thrived”-> native?

-line 40: gardens( -> gardens (

-line 41: you already said “holly genus”, so better say “Aquifoliaceae family”

-line 44-45: check citation style

-line 46: evolving into -> would be better “emerging as”

-line 47-48: the sentence of greenhouse effect, climate change factors and climate change, could be rewritten for smoothness. Also, is quite long, so do this change:

“regions. Drought, flooding, overheating, and extreme cold, may negatively impact the yield and quality of I. verticillata, thereby affecting the economic benefits of related industries, affecting the economic traits of its cut flowers (Parrotta et al., 2023)”

-line 52: “hence” without captions

-line 53-56: seems that this sentence needs a missing reference

-line 60: citation format

-line 61: I find difficult to understand this sentence “exhibits remarkable…its negative effects” What are those small-scale migrations within plantations? Please explain

-line 71: points = occurrences

-line 72: species(Elith et al., 2011).For -> species (Elith et al., 2011). For

-line 72-74: These examples are in China I guess, so mention that. If not, better to use only examples in China to by more robust with the statement.

-line 76: delete “model”

-line 77: “MaxEnt, in comparison to other models..”

-line 82: why here you mention climate-ecological models? That should be explained before in bioclimatic models, otherwise readers can start to confusing with different terms (even though they are the same)

-line 85-87: the sentence “its worth noting that…” needs more background. A quick search in Google Scholar shows that several articles have indeed validated MaxEnt predicitons. Maybe to improve the sentence you can say that few have done it using additional fieldwork data, or only few in Asia/China. So, in that way you are more precise.

-line 87-89: the first sentence you specify for “tree”, but then you comeback to be broader using “plants”. Also, your species is a shrub, so use that word, and change the order from macro to micro.

-line 90-91: “important indicator” of what? Seems that you have missing words.

-line 91-92: “different climate soil…” this sentence seems to not be in the correct part, as you first started with fruit traits, then soil, and then you comeback to fruits.

-line 94: What is an “excellent ecological value”?. Maybe sounds better “high ecological value” or “important ecological value”

-line 95-97: rewrite because sounds as a circular argument: like “because of scattered distribution, we studied its distribution, and we found was scattered”

-line 97-111: All these sentences are really related to the methods, so improve the streamlining or just move it to the methods section to move quicker to the aim of the study. Considering all the step that you have in the methods, a suggestion would be to have a workflow as a initial figure, so you can easily show to readers all the steps that you did.

Material and methods:

-line 116: why manifestation data? Why not just occurrences?

-line 117: dispersion -> distribution

-line 126: the reference for ENMTools is missing

-line 128: 2.5 x 2.5 arcmin

-line 131-134: why you needed the map of China? That was not clear. I guess it is for subtracting environmental data for the background envelope. Please explain

-Figure 1: “Specices” -> Specimens or species distribution

-general comment: so far is not clear for me is you pretend to do only one SDM at the global scale, or two, comparing native versus introduced area. If the first is the selecting, you need to explain why a comparison between the native and introduce area were not included

-line 137: most probably the high-quality image will still have too low resolution to see the triangles given their size. So, consider to change them to “dots”

-line 140: consider change it to “Climatic variables have shown to have important ecological significance for plants, affecting their distribution at broader scales”

-line 142: weather data -> “climatic data”. You used the biovariables so by the definition is climate (averages for 30 years period).

-line 142: Worldclim 2.1 or 1.4? (I guess is the first). Please include the version and the corresponding timeframe (1970-2000). You also need to discuss if the temporal frame of the biovariables correspond to most of your occurrence data. I guess that many I. verticilata in China have been introduced after the 2000’s, so they may not match properly the environmental data. That selection needs justification. For example, in this research says it was introduced since 2006 https://apsjournals.apsnet.org/doi/10.1094/PDIS-07-22-1688-PDN

-line 144: 2.5 arcmin

-line 144: you still do not know the potential future distribution, just selected the future climate scenarios, then should be “…of 2.5 arcmin. Future climate scenarios were selected…”

-line 146-150: improve the writing to avoid the overuse socioeconomic.

-general comment: I saw in the methods that you used R for correlations, for kuenm, and I guess that also for ENMTools, but you also used several other software such as ArcGis, SPSS 26 and Origin 95. I wonder why not do directly all in R to simplify and to be more replicable?

-Figure 2: is not cited in the manuscript

-line 162: which DEM resolution?

-line 181: check citation format

-line 207: is a bit awkward as it says that you collected fruits in sites without fruits. Rewrite the idea.

-line 208: tree or shrub?

-line 209: citation format

-line 212-213: seems to be a repeated idea

-line 228: after reading the whole document, I did not see any result about leaf traits, so better to exclude this sentence

-line 233: now you mention the reference period. This should be the first time that you talk about WorldClim

-line 238-243: the breaks still seem a bit arbitrary without explanation. Why these have different magnitude of interval? Here, I think that would be better for you to compare the global model vs and introduced model. The introduced model that will be more local, could behave quite differently, so would be wise to compare them. I suggest to review the methods of Gallien et al., 2012 https://onlinelibrary.wiley.com/doi/full/10.1111/j.1466-8238.2012.00768.x (specifically figure 3)

There you can have 4 different habitat suitability comparing both model with a threshold of 0.5, and more importantly for your study, with clear ecological – dispersion significance. For other examples, see fig 5a. of this research https://onlinelibrary.wiley.com/doi/full/10.1002/ece3.5295

Results:

-line 263: a good level of what?

-line 271: what do you mean with “covering the entire expanse in China”? As fig 3 shows that large part of the country is unsuitable

-line 294: I just wonder, these differences would all continue to express with a Tukey test (that is most robust). Would be nice to check.

-line 304: “varieties” -> “types”

-line 315-319: I had difficulties following the idea. Its look all the same to me, but the “In contrast” makes me feels that should not. Can you rewrite for clarity?

Line 320-330: I do not see the value of expressing all the results and all these percentages. It makes it really hard to follow. Try to streamline the idea with the most important results and interpretation. Readers should be able to read this in a figure/table. By the way, I do not know which figure I should read to follow all these percentages.

-Figure 7 & 8: I do not see them cited in the text. Fig. 8 seems to be important

-line 341: shrub or tree? Decide and keep constant

-line 346-348: improve writing. Avoid using three times “prediction”

-line 348-349: what are those “true species growth indicators”? Here the tricky word is “true” as is a difficult word to use in science.

-line 351-352: What do you mean with “high consistency”? It was not clear for me. Maybe you were you trying to say something like the field data (traits) was a novel way of validate the SDM?

-line 355: climate.The -> climate, the

Discussion:

-lines 360-381: you have a long discussion about soil variables, but these had a really low contribution in your model. Why you decide to give them such big importance?

-line 395: citation format

-line 442: citation format

-line 455: I. verticillate should be in italics

Reviewer #2: In the methodology it is important to add a GBIF doi of the data used. A way to get this doi in R is: "require(rgbif) require(dplyr)". It would be good to improve the resolution of figures 5 and 8, as the legends and scale are blurred. Have you evaluated the extrapolation of models in the methodology? This can be a critical point, the exclusion of adequacy predictions in environmental conditions considered extrapolative could be interesting to incorporate. See: https://nsojournals.onlinelibrary.wiley.com/doi/10.1111/ecog.06992

6. PLOS authors have the option to publish the peer review history of their article (what does this mean?). If published, this will include your full peer review and any attached files.

Reviewer #1: **Yes: **Dr. Nicolas Velasco

Reviewer #2: **Yes: **Emilly Layne Martins do Nascimento

---

## [Author Response · Author response to Decision Letter 0]

29 Jul 2024

Response to Reviewers

Responses to editorial comments:

1.The use of MaxEnt is widely accepted; however, modeling algorithms are very diverse and can produce different predictions. This becomes particularly important for non-native species and for estimates considering future climate change scenarios. There is a vast scientific literature recommending the use of different algorithms (e.g., MaxEnt, BIOCLIM, RandomForest among others). Therefore, I recommend that the authors use various modeling techniques and produce a consensus map

Response: According to your suggestion, I have added a model comparison table in the process of writing. In addition to the Maximum Entropy Model (MaxEnt), the Generalised Linear Model (GLM), the BIOCLIM model, and the Random Forest Model (RF) were also added to compare with the MaxEnt model in this paper, but according to the results obtained, the MaxEnt model is still the best. The modified MaxEnt model was finally adopted for the subsequent prediction.

2.The species “Ilex verticillata” originates from the Northeast of the United States and has already been introduced in various parts of the world. This was briefly mentioned by the authors. The critical point is which points were used for modeling? Note that this was also not clear to reviewer 1. According to the text and figure 1, the authors used points from the native region (USA) and China. By doing this, the authors are assuming that the species' niche is in equilibrium. Another option is to use only the points from the native area and project the species' distribution in the invaded area. This approach assumes that the niche is conserved. There is a vast literature discussing the transferability of niche models for the case of invasive species, and the authors need to justify the methodology used.

Response: Thank you for your suggestion, the actual points used for modelling are the points of occurrence within China, which have been modified in the text.

3.Please note that it is necessary to use the International Code of Nomenclature for algae, fungi, and plants when citing a plant species. For example, is necessary inform the taxonomic group (e.g. family) and author of the species.

Response: thanks for the suggestion, I have changed Ilex verticillata to Ilex verticillata (L.) A.Gray cies.

Response to reviewer 1's comments

1. My general impression is that the methodological section sounds logical but the overall manuscript will benefit from a better streamlining (specially a workflow figure at the beginning of the methods), and a consistency of terms. I have a major suggestion of how to improve part of the methods, specially the selection of suitability types, by using a comparison of an introduced model (China) vs a global one. However, I will understand if these changes are not totally feasible as you authors have already selected the 12 sites to test with field data. Other revisions are needed in typos, format, and discussion of soil variables. All in all, I had a great time reviewing the article so I will be glad of reviewing it again after the revisions.

Response: Many thanks to the reviewers for their meticulous revisions and suggestions, which I have benefited from during the revision process. Workflow diagrams have been added at the methodology according to your suggestions. The consistency of terminology I have revised as much as possible. Your suggestion to compare the introductory model (China) with the global model has not been done this time because the purpose of this paper is mainly the distribution of suitable areas in China and field trials have already been carried out, for which I am sorry. Other corrections such as typos and formatting have been made in the text. The excessive discussion of soil variables has been reduced according to your comments.

Response to reviewer 2's comments

1.In the methodology it is important to add a GBIF doi of the data used. A way to get this doi in R is: "require(rgbif) require(dplyr)". It would be good to improve the resolution of figures 5 and 8, as the legends and scale are blurred. Have you evaluated the extrapolation of models in the methodology? This can be a critical point, the exclusion of adequacy predictions in environmental conditions considered extrapolative could be interesting to incorporate. See: https://nsojournals.onlinelibrary.wiley.com/doi/10.1111/ecog.06992

Response: Thank you for your comments, I have added the doi of GBIF to the reference, the resolution of all the images in the paper has been upgraded to 300 dpi, due to the compression of the document for the images, so it will lead to blurring, and then I will add all the images to the compressed file to upload them individually. I have read the references you have provided, but this paper does not evaluate the model extrapolation method in the methodology, the Shape method calculates the degree of extrapolation given the multivariate distance from the projected data points to the nearest training data points, which is not in line with the prediction method used in this paper, and if changes are made it will increase the length and change too much, but thank you for the inspiration and advice, I will be writing my future papers using this method.

---

## [Decision Letter · Decision Letter 1]

3 Sep 2024

PONE-D-24-15741R1Predicting the Habitat Suitability of Ilex verticillata in China with Field-Test ValidationsPLOS ONE

Dear Dr. yin,

Thank you for submitting your manuscript to PLOS ONE. After careful consideration, we feel that it has merit but does not fully meet PLOS ONE’s publication criteria as it currently stands. Therefore, we invite you to submit a revised version of the manuscript that addresses the points raised during the review process.

Thank you for accepting the suggestions from both myself and the reviewers. In this new version, one reviewer has accepted the article while another has provided additional comments, necessitating a major revision. I also agree with the reviewer and indeed, some adjustments to the manuscript are still required. In addition to the reviewer’s comments, I also recommend the following:

1 - In the title, it is important to indicate the family Aquifoliaceae after the species name. This is a rule of botanical nomenclature, which I had previously commented on/suggested in the first version.

2 - A suggestion: The introduction consists of two long paragraphs. If the authors agree, it might be beneficial to divide them into smaller paragraphs. This would improve the readability of the manuscript.

3 - Reserve the last paragraph of the introduction to presented the objectives of the study.

4 - In the caption of Figure 1, remove the word "global." The figure highlights the distribution of points in China only.

5 - Line 130, merge this sentence with the previous paragraph.

6 - Line 170, it is mentioned that the models were generated for North America. Why? I believe the models were generated for China.

7 - Line 202, the authors indicated 12 sites to assess model accuracy. It would be helpful to show these 12 sites on a map. This could be done in Figure 1 along with the points used in the model. Simply indicate the 12 sites in different colors. I noticed that the coordinates were provided in Table 2, but the figure might present this information more clearly.

8 - Line 214: Were the assumptions of ANOVA tested? ANOVA is only cited in the table. It is important to include this information in the methods section of the text.

We look forward to receiving your revised manuscript.

Kind regards,

João Carlos Nabout

Academic Editor

PLOS ONE

Reviewers' comments:

Reviewer's Responses to Questions

**Comments to the Author**

1. If the authors have adequately addressed your comments raised in a previous round of review and you feel that this manuscript is now acceptable for publication, you may indicate that here to bypass the “Comments to the Author” section, enter your conflict of interest statement in the “Confidential to Editor” section, and submit your "Accept" recommendation.

Reviewer #1: (No Response)

Reviewer #2: All comments have been addressed

2. Is the manuscript technically sound, and do the data support the conclusions?

Reviewer #1: Yes

Reviewer #2: Yes

3. Has the statistical analysis been performed appropriately and rigorously? 

Reviewer #1: Yes

Reviewer #2: Yes

4. Have the authors made all data underlying the findings in their manuscript fully available?

Reviewer #1: Yes

Reviewer #2: Yes

5. Is the manuscript presented in an intelligible fashion and written in standard English?

Reviewer #1: Yes

Reviewer #2: Yes

6. Review Comments to the Author

Reviewer #1: Thanks for addressing most of my previous comments. Here I have some additional recommendations to polish further the manuscript

Line 112: “Global Biodiversity Information Facility”

Line 129: Figure 1. The chinese distribution points of I. verticillata.

Line 142: check the formats of the socioeconomic pathways � SSP1-2.6, SSP2-4.5, SSP5-8.5

Line 172: you should include a short explanation of this coefficients to select the models. What they tell to the reader? For example, is the first time that I see the KAPPA. Additionally, in Table 1 how the reader can discriminate between the different values? For example, RF values are very similar to MaxEnt. Is intuitive that you discriminate through higher values, but if a model has only a thousandth higher than other, that would be sufficient?

Line 256: I’m a bit confused here. I believe you need to mention earlier that you selected 15 variables, which were then used in the SDMs comparison. Some of this information should be moved to before line 242.

Line 264: This line is confusing to me for two reasons. First, when you mention “models using the first four climate variables,” are you implying that you tested models with different sets of variables? This is a common approach in optimization, but in the methods section, I didn’t catch that different sets of climate variables were tested. Second, I’m unclear about where to find the statement that these models “gave better results (AUC > 0.8)” or what they are being compared against.

Line 291: there is no legend

Line 296-299: SSP5-8.5

Line 307: Given that you have many figures, I suggest moving Figure 7 to the supplementary materials.

Fig 6: can also be a supplemental

Line 313: check SSP format from here onwards, and also in figure 6, 7, 8 and 9

Line 316: add a separation between the dot and However

Figures 6 & 9: Considering their size and resolution, these figures could benefit from removing the empty columns and rows separating the nine subgraphs.

Figure 8: Since there are significant differences between the types of suitability (making it hard to see differences in the categories with better suitability), it might be better to use a logarithmic scale for the y-axis.

Line 361: I do not understand what you mean here “two types of variables were included in the identical model”. What is the identical model?

Line 385: “sufficient” without capital

Line 451-455: is quite long. Try to separate the idea in two or three sentences

Reviewer #2: After analysing the revisions and corrections made by the authors in response to the suggestions and recommendations, I conclude that all the proposed changes have been duly implemented. The article now presents significant improvements in terms of clarity, consistency and scientific rigour for publication. The methodological and theoretical aspects have been adequately addressed, which reinforces the quality and relevance of the results presented. I therefore recommend that the article be accepted for publication, considering that it is capable of making a significant contribution to the scientific community.

7. PLOS authors have the option to publish the peer review history of their article (what does this mean?). If published, this will include your full peer review and any attached files.

Reviewer #1: **Yes: **Dr. Nicolás Velasco

Reviewer #2: **Yes: **Emilly Layne Martins do Nascimento

---

## [Author Response · Author response to Decision Letter 1]

3 Nov 2024

Response to Reviewers

Responses to editorial comments:

1 - In the title, it is important to indicate the family Aquifoliaceae after the species name. This is a rule of botanical nomenclature, which I had previously commented on/suggested in the first version.

Response: Sorry editor, in the last revision I changed the Latin name in the introduction but not the title, this time I added the family name after both the name and the species name in the introduction, I apologise for my oversight.

2 - A suggestion: The introduction consists of two long paragraphs. If the authors agree, it might be beneficial to divide them into smaller paragraphs. This would improve the readability of the manuscript.

Response: Already following your suggestion I have divided the two long paragraphs into 5 smaller ones, which I hope will increase the readability.

3 - Reserve the last paragraph of the introduction to presented the objectives of the study.

Response: The research objectives in the last paragraph of the introduction have been retained and divided into new paragraphs.

4 - In the caption of Figure 1, remove the word "global." The figure highlights the distribution of points in China only.

Response: In the description of Figure 1, it has been corrected to read The Chinese distribution points of I. verticillata.

5 - Line 130, merge this sentence with the previous paragraph.

Response: Line 130, this sentence has been merged with the previous paragraph.

6 - Line 170, it is mentioned that the models were generated for North America. Why? I believe the models were generated for China.

Response: Line 170, this problem was due to a translation error and has been corrected.

7 - Line 202, the authors indicated 12 sites to assess model accuracy. It would be helpful to show these 12 sites on a map. This could be done in Figure 1 along with the points used in the model. Simply indicate the 12 sites in different colors. I noticed that the coordinates were provided in Table 2, but the figure might present this information more clearly.

Response: Line 202, thanks to your suggestion, I have labelled the 12 sites in Fig. 1, using different colours to distinguish them.

8 - Line 214: Were the assumptions of ANOVA tested? ANOVA is only cited in the table. It is important to include this information in the methods section of the text.

Response: The assumptions of the ANOVA have been tested for p-values during the calculations. This information has been added additionally in the methods section of the paragraph.

Response to reviewer's comments:

I am glad to receive your comments again, not only it helps me a lot to revise my thesis this time, but also it is very inspiring for my future thesis writing, thank you very much.

Line 112: “Global Biodiversity Information Facility”

Response: Thank you, I have corrected this issue.

Line 129: Figure 1. The chinese distribution points of I. verticillata.

Response: I have corrected this to ’The chinese distribution points of I. verticillata.’

Line 142: check the formats of the socioeconomic pathways � SSP1-2.6, SSP2-4.5, SSP5-8.5

Response: Checks have been made and changes made throughout the text.

Line 172: you should include a short explanation of this coefficients to select the models. What they tell to the reader? For example, is the first time that I see the KAPPA. Additionally, in Table 1 how the reader can discriminate between the different values? For example, RF values are very similar to MaxEnt. Is intuitive that you discriminate through higher values, but if a model has only a thousandth higher than other, that would be sufficient?

Response: Well, the three types of indicators, KAPPA, AUC, and TSS, have different meanings and applications in their respective fields, and have been explained and supplemented in the text for the meaning of these three indicators.

Line 256: I’m a bit confused here. I believe you need to mention earlier that you selected 15 variables, which were then used in the SDMs comparison. Some of this information should be moved to before line 242.

Response: OK, I have added 15 variables related to selection in section 3.1 (formerly line 242).

Line 264: This line is confusing to me for two reasons. First, when you mention “models using the first four climate variables,” are you implying that you tested models with different sets of variables? This is a common approach in optimization, but in the methods section, I didn’t catch that different sets of climate variables were tested. Second, I’m unclear about where to find the statement that these models “gave better results (AUC > 0.8)” or what they are being compared against.

Response: In the process of model optimisation, I did test the model using different sets of variables and that step is outlined in the optimisation process, it was an oversight on my part that this was not mentioned in the methods section, it has now been added in the model optimisation methods section, hopefully this will answer your confusion.

Line 291: there is no legend

Response: Thanks for the heads up, I've added the image captions.

Line 296-299: SSP5-8.5

Response: The formatting issues in the content of lines 296-299 have been corrected.

Line 307: Given that you have many figures, I suggest moving Figure 7 to the supplementary materials. Fig 6: can also be a supplemental

Response: OK, I have given full consideration to your idea, but the text needs to cite Figures 6 and 7, so I have not put in the supplementary material for ease of reading.

Line 313: check SSP format from here onwards, and also in figure 6, 7, 8 and 9

Response: OK, the full SSP formatting has been revised.

Line 316: add a separation between the dot and However

Response: Thanks, I've added a separator between ‘point’ and ‘however’.

Figures 6 & 9: Considering their size and resolution, these figures could benefit from removing the empty columns and rows separating the nine subgraphs.

Response: Thanks for the heads up, I'll adjust the quality of the puzzle pieces to make sure the pictures are clear.

Figure 8: Since there are significant differences between the types of suitability (making it hard to see differences in the categories with better suitability), it might be better to use a logarithmic scale for the y-axis.

Response: Thank you very much for your suggestion, I have changed the y-axis to a logarithmic scale and hopefully the presentation is more intuitive.

Line 361: I do not understand what you mean here “two types of variables were included in the identical model”. What is the identical model?

Response: Two types of variables means two types of soil and climate variables, which I have modified.

Line 385: “sufficient” without capital

Response: Thanks for the correction, have changed the capital letters to lower case.

Line 451-455: is quite long. Try to separate the idea in two or three sentences

Response: Thanks for the suggestion, I have adjusted the statement to split the paragraph into two sentences.

---

## [Decision Letter · Decision Letter 2]

4 Dec 2024

Predicting the Habitat Suitability of Ilex verticillata (Aquifoliaceae)  in China with Field-Test Validations

PONE-D-24-15741R2

Dear Dr. yin,

We’re pleased to inform you that your manuscript has been judged scientifically suitable for publication and will be formally accepted for publication once it meets all outstanding technical requirements.

Kind regards,

João Carlos Nabout

Academic Editor

PLOS ONE

Additional Editor Comments (optional):

Reviewers' comments:

Reviewer's Responses to Questions

**Comments to the Author**

1. If the authors have adequately addressed your comments raised in a previous round of review and you feel that this manuscript is now acceptable for publication, you may indicate that here to bypass the “Comments to the Author” section, enter your conflict of interest statement in the “Confidential to Editor” section, and submit your "Accept" recommendation.

Reviewer #1: All comments have been addressed

2. Is the manuscript technically sound, and do the data support the conclusions?

Reviewer #1: (No Response)

3. Has the statistical analysis been performed appropriately and rigorously? 

Reviewer #1: (No Response)

4. Have the authors made all data underlying the findings in their manuscript fully available?

Reviewer #1: (No Response)

5. Is the manuscript presented in an intelligible fashion and written in standard English?

Reviewer #1: (No Response)

6. Review Comments to the Author

Reviewer #1: (No Response)

7. PLOS authors have the option to publish the peer review history of their article (what does this mean?). If published, this will include your full peer review and any attached files.

Reviewer #1: **Yes: **Dr. Nicolás Velasco

---

## [Editor Report · Acceptance letter]

17 Dec 2024

PONE-D-24-15741R2 

PLOS ONE

Dear Dr. yin, 

I'm pleased to inform you that your manuscript has been deemed suitable for publication in PLOS ONE. Congratulations! Your manuscript is now being handed over to our production team.

Kind regards, 

on behalf of

Dr. João Carlos Nabout 

Academic Editor

PLOS ONE